# Wooden Solar Evaporator Design Based on the Water Transpiration Principle of Trees

**DOI:** 10.3390/ma16041628

**Published:** 2023-02-15

**Authors:** Wei Xiong, Dagang Li

**Affiliations:** 1College of Materials Science and Engineering, Nanjing Forestry University, Nanjing 210037, China; 2College of Landscape Architecture, Jiangsu Vocational College of Agriculture and Forestry, Jurong 212400, China

**Keywords:** water transpiration, double-sided carbonization, poplar, wooden solar evaporator, solar interface, evaporative power

## Abstract

The double-sided carbonization of poplar with different sections forms a three-layer structure inspired by tree water transpiration. A photothermal evaporation comparison experiment was conducted to simulate the influence of solar radiation intensity (1 kW·m^−2^) on uncarbonized and single- and double-sided carbonized poplar specimens. The tissue structure, chemical functional group changes, and profile density of the specimens were analyzed using scanning electron microscopy, Fourier transform infrared spectrometry, and X-ray profile density testing, respectively. The results showed that the tissue structure of the specimen changed after treatment, and the relationship of water evaporation was shown as follows: cross-section (C) > Radial section (R) > Tangential section (T), and Double-sided carbonized poplar (DCP) > Single-sided carbonized poplar (SCP) > Non-carbonized poplar (NCP). Of these, the maximum photothermal evaporation was from the cross-section of the double-sided carbonized poplar (NCPC) with a value of 1.32 kg·m^−2^·h^−1^, which was 21.97% higher than single-sided carbonized poplar (SCPC) and 37.88% higher than non-carbonized poplar (NCPC). Based on the results, double-sided carbonization three-layer structure treatment can improve the evaporation force of the poplar interface, thereby improving the moisture migration ability of wood, and can be applied to solar interface absorber materials.

## 1. Introduction

The system of traditional solar water purification has high requirements on infrastructure, investment, and land area, which increases the difficulty of large-scale promotion in water-scarce areas [1]. Solar interface water vapor evaporation is a new technology that utilizes solar energy to form a liquid-solid-vapor interface on an absorber material to achieve efficient water purification [2,3,4]. This technology has a good application value for alleviating water pollution and improving water quality [5]. Currently, most studies focus on the selection of absorber materials, photothermal management, and water transport efficiency [6,7,8].

Absorber materials in solar interfacial water vapor evaporation technology are commonly based on plasmonic- and carbon-based materials [9]. In terms of plasmonic materials, gold nanoparticles [10], silver nanoparticles [11], and transition-metal nitrides [12,13] have been applied, with photothermal evaporation capacities of 1.39 kg·m^−2^·h^−1^ [11], 1.59 kg·m^−2^·h^−1^ [14], or even 2.45 kg·m^−2^·h^−1^ [15]. However, in contrast to the complex preparation process and high cost of plasmonic materials, carbon-based materials have low cost, good light absorption, and stability, and are one of the main absorbing materials for solar photo-steam conversion [16].

As a natural carbon-based material, wood has become the research object of solar interface photothermal evaporation absorber materials because of its good hydrophilicity, abundant, crisscrossed micro-nano channels, and good floatability on the water surface [17,18]. To obtain high-efficiency water vapor evaporation performance, many scholars have developed reasonable micro-nano structure designs for wood, such as the overall modification of natural surface wood by plasma [19], three-dimensional graphene [20,21], carbon nanotubes along the growth direction [22], and carbonization treatment [23,24]. Optimizing the one-dimensional nano-water transport channel and three-dimensional absorber structure provides efficient channels for water and steam transport greatly reduces the radiation and convective heat losses, and yields a high-energy conversion efficiency. However, most of these studies were of the single-sided modification of wood, and only a two-layer structure was formed. Under one solar radiation intensity (1 kw·m^−2^), the photothermal evaporation ranged from 0.95 to 1.85 kg·m^−2^·h^−1^ [25,26]. Recently, 2.70 kg·m^−2^·h^−1^ evaporation has been reported [27], which is one of the research directions of chemical treatment of the structure of the wood to create a conduit channel to improve water vapor migration ability.

Nature has already provided an excellent solution for efficient evaporation: long-distance water transport by roots, trunk, and leaves of trees [28,29,30]. Specifically, the low water potential (tension) generated by the transpiration of the leaf surface provides a suction force that gradually transfers the tension to the roots through the continuous water column in the xylem, resulting in a water potential on the root surface that can absorb water from the soil and lift the water to the crown. However, how can non-living wood enhance its water absorption and migration abilities? In this study, inspired by tree water transpiration, fast-growing poplar with straight fiber, simple structure, and wide range of cultivation was selected as the experimental material and a double-sided carbonization (carbonized thickness 2 mm) of poplar was developed to form a three-layer structure to simulate the root, trunk, and leaf structures of a tree. The bottom of the specimen is carbonized to form a fine porous liquid-solid interface like root hair, which is convenient for water adsorption; and the top of the specimen is carbonized to form a photothermal energy storage layer, while the liquid-vapor interface is realized under a solar radiation intensity (1 kw·m^−2^), resulting in moisture concentration gradient and temperature difference, providing the wood with bottom-up moisture lifting force. Through the photo-thermal evaporation contrast test on the non-carbonized, single-sided carbonized, and double-sided carbonized poplar specimens with different sections, the microstructure, chemical functional group changes, and section density of the specimens were analyzed by scanning electron microscopy, Fourier transform infrared spectroscopy and X-ray section density testing, and the influence of different sections and carbonization treatments on the improvement of the water migration ability of poplar was discussed.

## 2. Materials and Methods

### 2.1. Materials

Poplar wood: Populus × canadensis ‘I-214’, a 12-year-old growing season poplar, from Suqian, Jiangsu. Its diameter at breast height was approximately 400 mm; the sapwood part was used, and the width of the growth ring was approximately 12–15 mm. Through natural drying, the air-dried wood density was 0.52 kg/m^3^, and the dimensions of the specimen were R × T × L, R × L × T, and T × L × R (30 mm × 30 mm × 10 mm).

Vacuum pre-drying:, the poplar specimens of each section were vacuum-dried to avoid the deformation of the wood block during carbonization in the following conditions: temperature T = 80 °C, pressure P = −0.08 MPa, time t = 12 h.

Surface carbonization: A closed electric furnace (Shanghai Lichen Technology, Shanghai, China) was set at 400 °C to perform surface carbonization (for cross, tangential, and radial). Considering the wood ignition point factor (280–300 °C), a combination of short-term carbonization (400 °C) and water cooling (20 °C) was adopted to ensure that the carbonization interface was clear. The carbonization layer was controlled within a range of 2 mm of a pre-drawn line, and single- and double-sided carbonization was performed, respectively, as shown in Figure 1.

After the previously mentioned treatments, five uncarbonized, single-sided carbonized, and double-sided carbonized specimens were prepared. The untreated poplar was labeled NCP, the single-sided carbonized poplar SCP, the double-sided carbonized poplar DCP, and the fully carbonized poplar as FCP; L, C, R, and T represent the longitudinal, cross, radial, and tangential sections, respectively.

### 2.2. Methods

#### 2.2.1. Photothermal Evaporation Test

A photothermal evaporation system, as shown in Figure 2a,b, was designed to verify the effect of photothermal evaporation utilizing the uncarbonized and single- and double-sided carbonized poplar on water migration,. The indoor temperature and humidity were controlled and set at 20 °C and 40%, respectively. A temperature and humidity meter was used for real-time monitoring. As shown in Figure 2c, the test piece was inserted into the square hole of the PVC board according to the actual size and flushed with the surface of the PVC board. Only the floating surface had static contact with water, and the side of the specimen, treated with sealing wax, had no contact with the water surface. A photochemical atmospheric pressure reactor (CEL-HXF300-T3, Beijing China Education Au-light Co., Ltd., Beijing, China) simulates the solar light source and is adjusted to the standard solar radiation intensity of 1 kW m^−2^ (optical power meter: CEL-NP2000-2 (10), Beijing China Education Au-light Co., Ltd., Beijing, China) to irradiate the poplar specimen to form a gas-liquid evaporation interface. An electronic balance (Sartorius 0.1 mg, Sartorius, Göttingen, Germany) was used to record data every 5 min to accurately measure the change in the evaporative mass. A FLIR thermal imager (FLIR, Wilsonville, OR, USA) was used to record a thermal image of the evaporation interface every 5 min to characterize the photothermal intensity of the evaporation interface.

#### 2.2.2. Microstructural Characterization Test

A scanning electron microscope (FEI, Hillsboro, OH, USA) was used to observe the microscopic characterization of the poplar specimens and understand the effect of carbonization treatment on the poplar cell channels.

#### 2.2.3. Fourier Transform Infrared Spectroscopy

The chemical functional groups on the surface of the specimens were determined using Fourier transform infrared transform (FTIR) spectroscopy (Bruker Tensor27, Bruker Corporation, Billerica, MA, USA) analysis. Spectral scanning was performed eight times, with a resolution of 4 cm^−1^, and the acquisition range of the infrared spectrum was set at 500–4000 cm^−1^ to understand the changes in the chemical functional groups on the surface of the poplar wood after carbonization.

#### 2.2.4. X-ray Profile Density Test

The X-ray profile density test of the specimen was conducted using an X-ray Lab Density Gauge (Electronic Wood Systems GmbH, Hameln, Germany). The specimen is the cross-section of samples of dimensions R × T × L = 50 × 50 × 10 mm prepared using the same carbonization treatments as above. The uncarbonized and single- and double-sided carbonized specimens were tested using an X-ray profile density analyzer to understand the change in the vertical section density of the poplar after the various treatments.

## 3. Results

### 3.1. Estimation of Water Quality Change and Evaporation Force in Photothermal Evaporation

Figure 3a shows that the change in the evaporative water quality of untreated poplar is NCPC > NCPR > NCPT, which conforms to the law of water migration in the wood of all directions, that is longitudinal > radial > tangential. By comparing the photothermal evaporation data under the same conditions for the three sections, Figure 3b shows that the cross direction is DCPC > SCPC > NCPC. As shown in Figure 3c,d, the quality change of evaporative water on the radial and tangential sections is DCPR > SCPR > NCPR and DCPT > SCPT > NCPT, respectively. This shows that surface carbonization treatment has a greater effect on water transfer performance than non-carbonization treatment, and the effect of double-sided carbonization is better than that of single-sided carbonization. Among them, the maximum evaporation of the cross-section double-sided carbonized poplar (NCPC) is 1.32 kg·m^−2^·h^−1^, which is 21.97% higher than single-sided carbonized poplar (SCPC) and 37.88% higher than non-carbonized poplar (NCPC).

An increase in temperature can improve the diffusion strength of water vapor in wood and reduce the viscosity of liquid water allowing the water to flow faster along various conduction paths and, thereby, water migration is promoted. Figure 4a shows the results of a photothermal transpiration test with 1 kW m^−2^ radiation intensity with the NCPC, SCPC, and DCPC specimens under the same air-drying conditions. An infrared thermal imager was used to record the average temperature changes of the surface of each specimen after 0, 15, 30, 45, and 60 min of illumination. As Figure 4c shows, the surface temperatures of the SCPC and DCPC specimens were much higher than that of NCPC. This is because the surface layer of the poplar forms a black heat-gathering effect owing to carbonization, and its light absorption performance was higher than that of NCPC. The photothermal evaporation effect of the top energy-storage layer, resulting in moisture gradients and temperature differences, provides bottom-up moisture lifting power for the wood, resulting in higher water evaporation by the SCPC and DCPC specimens than that by the NCPC. After the carbonization treatment, the thermal imaging temperature of the DCPC specimen was lower than that of the SCPC. This is because the water absorption layer at the bottom of DCPC is fine pores like tree root hairs, which significantly improve the water absorption capacity. Compared with SCPC, the water evaporation of DCPC is larger, and its temperature is about 1 °C lower. This is also confirmed by the water evaporation of each specimen after 60 min: NCPC, 0.82 kg·m^−2^; SCPC, 1.03 kg·m^−2^; DCPC, 1.32 kg·m^−2^, showing a relationship of DCPC > SCPC > NCPC.

Through the photothermal evaporation test, the evaporation force (*E*_0_) of water migration can be calculated. In the evaporation test, as the surface moisture of the specimen increased, when the moisture content was higher than the fiber saturation point, the evaporation of the specimen (M) under atmospheric pressure could be approximately calculated using Dalton’s formula [31]:(1)$$M=B\;\left(P_{H}-P_{n}\right)$$
where $\left(P_{H}-P_{n}\right)$ is the evaporation force $E_{0}$, $P_{H}$ is the water vapor partial pressure on the surface of the material, $P_{n}$ is the water vapor partial pressure of the surrounding air, and B is the evaporation coefficient.

When the airflow direction was parallel, and the temperature of the evaporated water surface was 0–250 °C:(2)B=0.0017+0.0013 ω
where *ω* represents the airflow velocity. This test is in a closed state, and *ω* can be approximated as 0 m/s. As shown in Figure 4, the evaporation surface temperature was lower than 50 °C, from which the evaporation force *E*_0_ can be preliminarily calculated using the following equation:(3)$$E_{0}=\left(P_{H}-P_{n}\right)=\frac{M}{B}=\frac{M}{0.0017}$$

According to Equation (3) and the *M* value obtained through the photothermal transpiration test, it can be estimated that the evaporation forces of NCPC, SCPC, and DCPC at 60 min were 779.23, 608.55, and 483.30 mm H_2_O, respectively. Thus it still follows the relationship of DCPC > SCPC > NCPC, as shown in Figure 5.

### 3.2. Moisture Migration Microstructure Characterization Analysis

Moisture in wood mainly moves to the surface in the gaseous and liquid states, according to the three types of conduction paths. The first (Ⅰ) is the large capillary path, in which the moisture content gradient is formed inside the wood owing to the evaporation of moisture from the surface of the wood. Under the pressure of water vapor, water diffuses to the wood surface in a gaseous state along the adjacent cell cavities, pit cavities, and micropores on the pit membrane. The second (Ⅱ) is the microcapillary path, in which the bound water moves from the high-water to low-water content wood under the action of capillary tension. The third (Ⅲ) is the mixing path, in which the water continuously moves or diffuses along adjacent capillaries and cell cavities alternating between the liquid and gaseous states.

As shown in Figure 6, the scanning electron microscopy (SEM) test shows that poplar has typical hardwood tissue characteristics. The main constituent cells are vessels, wood rays, wood fibers, and axial parenchyma. Compared with most woods, poplar is a tree species with lightweight and developed pores, suitable as a solar interface photothermal evaporation absorber material. By comparison, it was found that many micropores and transition pores formed on the surface of the carbonized poplar, which had not only a high specific surface area but also a good adsorption performance after excluding the tar substances in the pores (CPC). The longitudinal pipes of poplar wood after carbonization were also smooth (CPR). The chordwise pit tissue was affected by high temperatures, and the pit membrane was damaged, making the lateral transport of water (CPT) easier.

As shown in Figure 7, the surface carbonization of the double-sided carbonized specimen can be regarded to represent the canopy layer, which provides energy storage, transpiration effect, negative-pressure water potential, and suction effect. The middle layer can be regarded to represent the tree trunk and acts as a water transport channel. The bottom carbonized layer, which is the water absorption layer, is the root of the tree. Thus the double-sided carbonized specimen forms a three-layer structure, similar to canopy transpiration, trunk water transport, and root water absorption.

In the double-sided carbonized poplar block, the carbonized layer of the specimen (c) (Figure 7c) is similar to the fibrous root of the plant (c’). The carbonization layer is in contact with the water surface, and the black carbon layer has a strong water absorption capacity, under which water mainly migrates in (Ⅱ) mode, and absorbs water quickly. The xylem layer of the specimen (b) (Figure 7b) is similar to the trunk of a plant (b’). The duct lifts the water to the top with capillary tension, and the water penetrates laterally between pits and transverse ray cells, under which water migration mainly occurred in (Ⅲ) mode. The carbonized layer of the specimen (a) (Figure 7a) is similar to the leaves of plants (a’). Because the black carbon layer absorbs heat, the water temperature at the top is increased, and a water temperature gradient is formed in the wood tissue from top to bottom, thus forming a water vapor pressure difference at the top, resulting in the specimen being prone to gaseous and liquid mass transfer, which enhances water transpiration. It continuously stores heat to enhance the transpiration cycle, under which water migration mainly occurred in (I) mode.

### 3.3. Profile Density and Composition Analysis

The chemical composition of the poplar cell wall consists of cellulose, hemicellulose, lignin, and small amounts of pectin, protein, extract, and ash. Invading bodies, starch granules, and other substances may be present in the cell cavities. Figure 8 shows the radial views of the NCPC, SCPC, and DCPC specimens from the perspective of their thickness (mm). Carbonized from the surface to the inside, its composition and structure density undergo a degree of change.

Figure 8a shows the FTIR analysis of non-carbonized (NCP) and fully carbonized poplar (FCP). Compared with NCP, FCP showed a large variation in the absorption peaks at 1732, 1610–1592, 1327, 1240, and 1109 cm^−1^ wave numbers. This is the C=O stretching vibration of the conjugate (hemicellulose lignin), C=C stretching vibration (lignin benzene ring), C–O stretching vibration (lignin lilacs structure), and C–H in-plane bending vibration (lignin lilacs structure). The effect of carbonization on lignin is greater, and these functional groups are located on the surface of the wood and greatly impact the physical and chemical properties of the surface. As shown in Figure 8b, in the X-ray profile density test, the profile density linearity of NCPC is straight and uniform, indicating that its internal structure is homogeneous. The 0–2 mm area of the SCPC in the thickness direction is non-carbonized, showing a relatively uniform profile density, followed by a downward trend after 2 mm and a sharp drop after 8 mm, indicating that the lignin loss is large after entering the carbonized layer, and the structural pores are large. The profile density of DCPC in the thickness direction shows an increasing-homogeneous maintaining-decreasing trend, indicating that after double-sided carbonization, the carbonized layers on the surface and bottom also lose lignin and are loose in structure with strong water migration properties.

## 4. Conclusions

The three-layer structure of double-sided carbonized poplar can better simulate the water transpiration mechanism of the roots, trunks, and leaves of trees. Driven by solar energy, the use of double-sided carbonized poplar significantly improved the water transpiration quality, and the water evaporation followed the trend of DCPC > SCPC > NCPC. Of these samples, NCPC had a photothermal evaporation capacity of 1.32 kg·m^−2^·h^−1^, which was 21.97% higher than SCPC and 37.88% higher than NCPC.Carbonized poplar had many micropores and transition pores on its surface, forming a high specific surface area. The tar material in the pores exhibited a good adsorption performance after exclusion. In addition, the longitudinal pipes were unobstructed; the pit tissue in the chord direction was affected by the high temperature, and the pit membrane was damaged, which made it easier for water to transmit laterally. Structural changes were carried out by wood physical treatment, but according to the existing literature, the transformation of the internal structure of the wood is an effective way to further improve the photothermal evaporation capacity.The evaporation force of the test piece was estimated by the experimental measurement of photothermal evaporation. This provides a certain reference value for estimating the water migration of xylem surfaces under photothermal conditions. In the future, the theory of molecular dynamics will be incorporated into the simulation analysis of the migration of water molecules in wood, which will have significance both on a microcosmic scale and in providing continuous guidance.As an absorber material, the mass of photothermal evaporation is also related to the tree species. In the photothermal evaporation test of a double-sided carbonized three-layer structure of spruce recently completed by the author, the maximum water evaporation was 0.94 kg·m^−2^·h^−1^, which was lower than the DCPC water evaporation of poplar (1.32 kg·m^−2^·h^−1^). This was due to the difference in the transporting molecules (tracheids and vessels) of the two kinds of wood. In the future, the selection of various kinds of wood for water purification is also a research direction.

## Figures and Tables

**Figure 1 materials-16-01628-f001:**
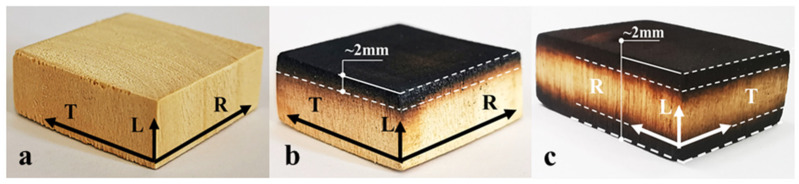
Processing diagram of the specimens. (**a**) Cross-section of untreated poplar; (**b**) cross-section of single-sided carbonized 2 mm poplar; (**c**) cross-section of double-sided carbonized 2 mm poplar.

**Figure 2 materials-16-01628-f002:**
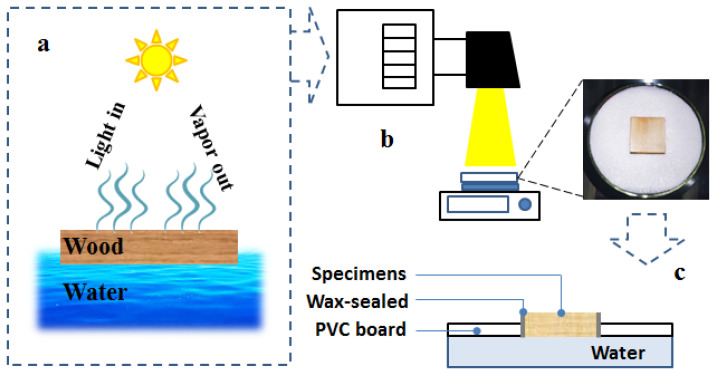
The schematic of the photothermal transpiration test. (**a**) Experimental design of a wooden solar evaporator; (**b**) test method diagram; (**c**) PVC (5 mm thickness) was made into a circular board of 90 mm diameter to match the diameter of the Petri dish. A square hole of the same size as the test piece was cut at the center of the PVC plate. The test piece was inserted into the square hole of the PVC circular board, and the lower surface of the test piece was flush with the lower surface of the PVC. The assembled PVC circular board floated on a Petri dish filled with pure water such that the surface of the test piece was in contact with the water.

**Figure 3 materials-16-01628-f003:**
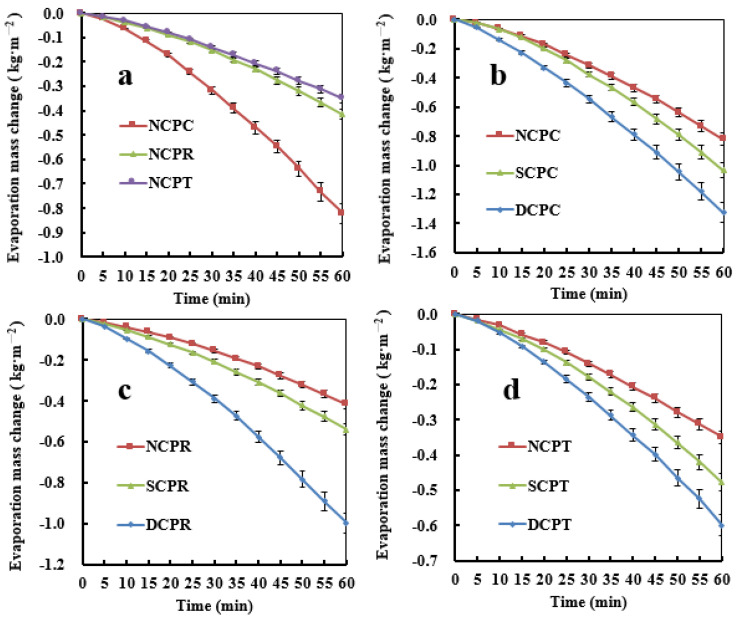
Changes in the quality of photoevaporation water of specimens under 1 kW·m^−2^ light intensity. (**a**) change in evaporation mass of untreated poplar in three sections; (**b**) change of evaporation mass in cross-section of specimens; (**c**) change in evaporation mass in the radial section of specimens; and (**d**) change in evaporation mass in the tangential section of specimens.

**Figure 4 materials-16-01628-f004:**
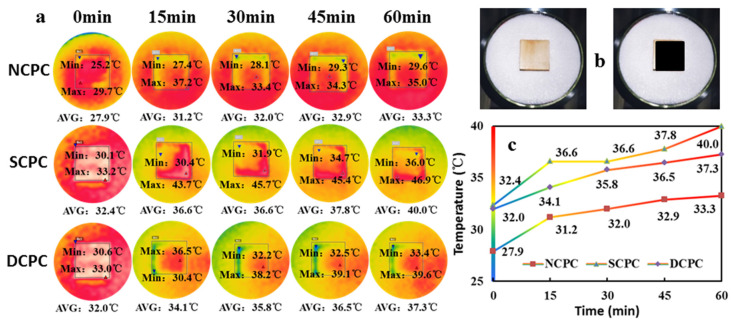
Infrared image of specimen at 1 kW·m^−2^ simulated solar illuminance. (**a**) Thermal imaging change diagram of specimens; (**b**) installation diagram of specimens; and (**c**) temperature variation diagram of specimens.

**Figure 5 materials-16-01628-f005:**
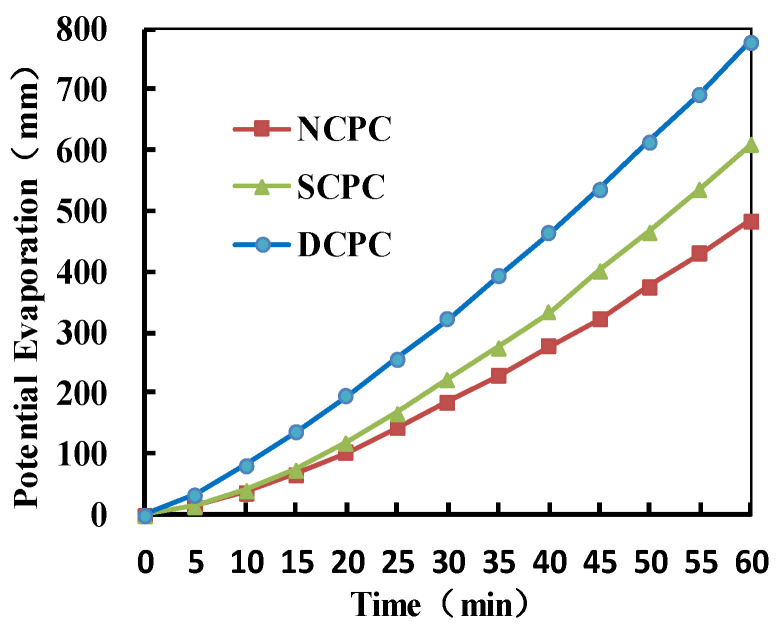
Evaporation force variation.

**Figure 6 materials-16-01628-f006:**
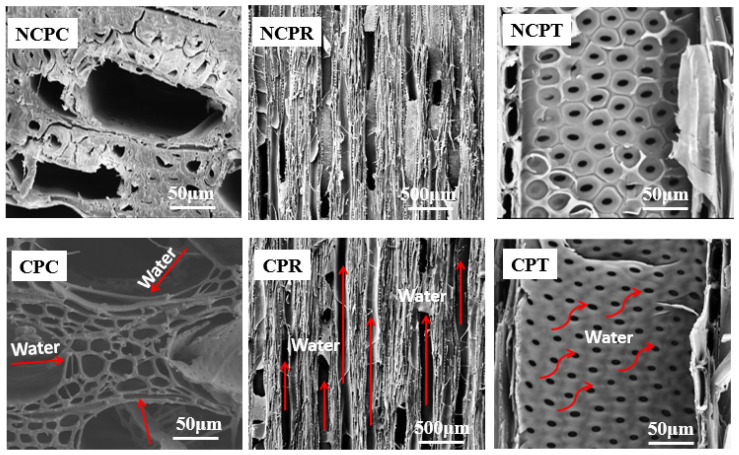
Cross-section, radial section, and tangential section structures of untreated poplar and carbonized surface poplar.

**Figure 7 materials-16-01628-f007:**
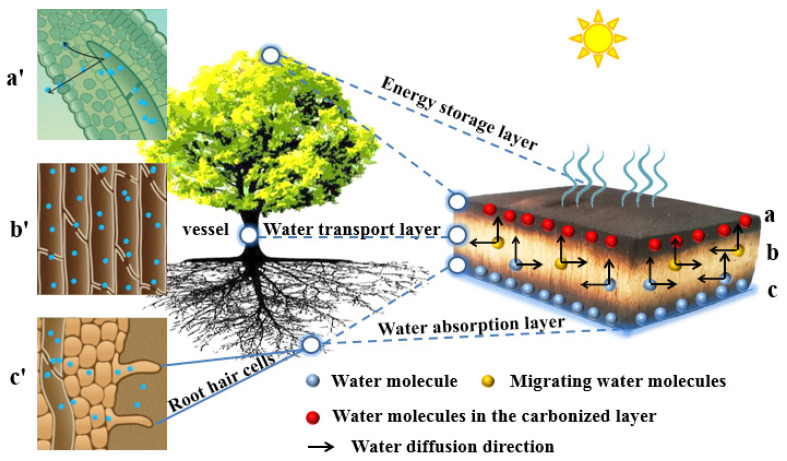
Water migration map of double-sided carbonized poplar specimens corresponding to plants. (**a**) The photothermal energy storage layer corresponding to the transpiration effect of plant leaves (**a’**); (**b**) the water transport layer corresponding to the water migration effect of plant ducts (**b’**); (**c**) the water absorption layer corresponding to the water absorption effect of tree roots (**c’**).

**Figure 8 materials-16-01628-f008:**
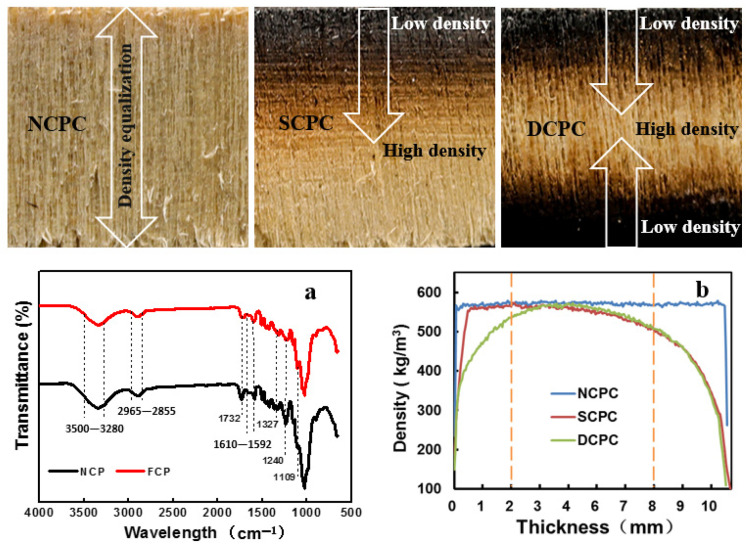
Profile density and composition change analysis. (**a**) FTIR analysis of uncarbonized and fully carbonized poplar; (**b**) sectional density analysis of NCPC, SCPC, and DCPC specimens from the perspective of their thickness (mm).

## Data Availability

Not applicable.

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
