# Peer review of "Wooden Solar Evaporator Design Based on the Water Transpiration Principle of Trees"

_materials, 2023, doi:10.3390/ma16041628_

Round 1
Reviewer 1 Report
The authors demonstrated the Wooden solar evaporator design based on the water transpiration principle of trees. It is an interesting field of research. Their claims have been supported by several experimental results. I thoroughly enjoyed the paper. The paper may accept in this journal, however, the authors may demonstrate if any hydrogen bonding solvent is used what will be the rate of evaporation.
Reviewer 2 Report
Dear Authors,
The paper presents interesting aspects about wooden solar evaporation.
Still several aspects need be solved:
1. The introduction does not explain very clear: what is the importance of poplar transpiration? why poplar and not another wood?
2. An explanation about the processes used in order to carbonize poplar is very useful. How much time? What temperature, etc. The images in Fig. 8 show different carbonization statuses. The carbonization seems to be different.
3. Please explain Fig. 3a. It is not clear what is presented. Is it the evaporation mass / minute / 5 minutes / cumulative mas … ?
4. The Y axis in Fig. 3a is noted as “evaporation mass change” and the caption mentions “Evaporation quality”. Please unify.
5. Fig. 6 shows single sided or double sided carbonized surface?
6. Line 209: “is similar to the diameter of a plant” in what? The similarity should be visible in one of the images / diagrams.
7. Line 211+212: the same remark as above.
8. In order to complete the analysis, Fig. 6 and 8 should be completed with all 9 types of samples presented.
9. Conclusions should be more detailed. They are very vague now (L249: “better”, L259 “certain”, etc.).
10. The results obtained should be compared with other studies, wood types etc.
11. Examples of applications for the results should be also mentioned.
I hope that my comments will be useful.
Sincerely,
Reviewer
Reviewer 3 Report
The manuscript is well written and the results are clearly presented. The conclusion is also supported by the measurements made. Carbonising the top surface enhances the evaporation, possibly because the blackened surface can absorb the radiation and increase the temperature/evaporation. The difference between single and double layers is, however, not very significant in some cases. Hence, the linkage to the tree analogy is very weak since the generation mechanism for the evaporation is very different.
One suggestion is for the authors to compare the evaporation rate with the many different materials reported in the literature. With the comparison, the readers can then better understand whether this approach can be used efficiently in considerations of the other alternatives.
Reviewer 4 Report
Authors in this manuscript used double sided carbonization process for poplar to enhance water evaporation processes.
Here are may comments on it:
1- The abstract needs more clarification regarding the results. I suggest adding quantified results like the percentage of enhancement for example.
2- In the introduction, line 24: "This technology has a good application....". I understand that this statement is cited however, it is a bit misleading to use the word "good". Please use a measurable expression or re-write the sentence.
3- The problem statement is not entirely clear in the introduction. Paragraph 3 in the intro discusses the pros, but what are the downsides? Please clarify your contribution here.
4- In section 2, how did you maintain a 2mm carbonized thickness? Was it time based? Please add the basis on which you achieved 2mm on wood.
5- There is tis problem with abbreviations: NCP, SCP, DCP....You need to add a nomenclature to the manuscript.
6- In section 4: NCPC, NCPR, NCPT...same thing. Add those to the nomenclature.
7- Figure 8-b shows that the density of DCPC is not symmetrical across the first and last 2mm. It is a slight asymmetry but brings my previous point (point 4) again. How do you make sure the carbonization is uniform?
Reviewer 5 Report
The manuscript entitled ‘Wooden solar evaporator design based on the water transpiration principle of trees’ presents an experimental investigation of poplar wood-based evaporator design using simulated solar flux. The submission needs major revision in the light of the following comments are to be addressed and major revision is required
1. The abstract must be supplemented with experimental findings in facts and figures. Moreover, the novelty of the work must be highlighted in the abstract.
2. The manuscript contains typos, grammatical and structural mistakes and should be proofread by a native English speaker.
3. The introduction section must be updated with the relevant literature, the authors may comment on related published works e.g. ‘Journal of the Brazilian Society of Mechanical Sciences and Engineering 44,: 427 (2022), ‘Energies 2020, 13(18), 4956’, etc. in the revised version of the manuscript. The last paragraph of the introduction section should highlight the novelty and a brief methodology should be added.
4. Combined references like [13-15], [16-18], etc. should be avoided.
5. Line 55, the statement “Populus × canadensis ‘I-214’, a 12-year-old growing season poplar, from Suqian, Jiangsu.” Should be improved.
6. Line 58, in “R×T×L”, R, T and L must be defined. What is the difference between R×T×L, R×L×T and T×L×R?
7. The text from line 62 to line 76 including Fig,1 should be a part of the section 2.2 instead of section 2.1.
8. Fig. 6 and Fig. 8 and their respective details should be shifted to sections 2.2.3 and 2.2.4, respectively.
9. Line 124, NCPC>NCPR>NCPT, and DCPC>SCPC>NCPC, the acronyms are not defined. It is advised to define the acronyms at their first appearance in the text.
10. A list of nomenclature should be added to include all the acronyms and the parameters of the equations along with their units.
11. Fig. 3, the legends are not defined anywhere in the manuscript.
12. Why the temperature for SCPC case in Fig. 4(c) is the same from 15min time to 30min time?
13. How was the evaporation in Fig. 5 measured?
14. The conclusion should be rewritten to include the facts and figures of the experimental findings.
15. A new section ‘Challenges and Prospects’ should be added just before the conclusion section to explain the challenges faced during this work and possible future prospects.

Round 2
Reviewer 2 Report
Dear Authors,
The paper was improved, but some aspects need be solved:
1. I did not find any novelty regarding point 10 and 11 in the previous review.
No comparison with other similar studies, wood types etc. (the word “spruce” doesn’t even appear once in the paper).
No examples of real applications for the results are mentioned.
2. Also, mentioning the efficiency of such a material would increase the quality of the paper. How many liters of water can be filtrated with such a specimen? What type of pollutants cane be eliminated from the water? …
I hope that my comments will be useful.
Sincerely,
Reviewer
Reviewer 5 Report
The authors have adequately addressed all of my observations and the revised manuscript is endorsed for publication.
Author Response
Dear teacher, thank you very much for your rigorous opinion!